# The Tricky Connection between Extracellular Vesicles and Mitochondria in Inflammatory-Related Diseases

**DOI:** 10.3390/ijms24098181

**Published:** 2023-05-03

**Authors:** Tommaso Di Mambro, Giulia Pellielo, Esther Densu Agyapong, Marianna Carinci, Diego Chianese, Carlotta Giorgi, Giampaolo Morciano, Simone Patergnani, Paolo Pinton, Alessandro Rimessi

**Affiliations:** 1Department of Medical Sciences, Section of Experimental Medicine, Laboratory for Technologies of Advanced Therapies, University of Ferrara, 44121 Ferrara, Italy; tommaso.dimambro@unife.it (T.D.M.); giulia.pellielo@unife.it (G.P.); estherdensu.agyapong@edu.unife.it (E.D.A.); marianna.carinci@unife.it (M.C.); diego.chianese@unife.it (D.C.); carlotta.giorgi@unife.it (C.G.); giampaolo.morciano@unife.it (G.M.); simone.patergnani@unife.it (S.P.); paolo.pinton@unife.it (P.P.); 2Center of Research for Innovative Therapies in Cystic Fibrosis, University of Ferrara, 44121 Ferrara, Italy

**Keywords:** mitochondria, extracellular vesicles, mitochondria-derived vesicles, mitovesicles, inflammation, intercellular communication, inflammatory diseases

## Abstract

Mitochondria are organelles present in almost all eukaryotic cells, where they represent the main site of energy production. Mitochondria are involved in several important cell processes, such as calcium homeostasis, OXPHOS, autophagy, and apoptosis. Moreover, they play a pivotal role also in inflammation through the inter-organelle and inter-cellular communications, mediated by the release of mitochondrial damage-associated molecular patterns (mtDAMPs). It is currently well-documented that in addition to traditional endocrine and paracrine communication, the cells converse via extracellular vesicles (EVs). These small membrane-bound particles are released from cells in the extracellular milieu under physio-pathological conditions. Importantly, EVs have gained much attention for their crucial role in inter-cellular communication, translating inflammatory signals into recipient cells. EVs cargo includes plasma membrane and endosomal proteins, but EVs also contain material from other cellular compartments, including mitochondria. Studies have shown that EVs may transport mitochondrial portions, proteins, and/or mtDAMPs to modulate the metabolic and inflammatory responses of recipient cells. Overall, the relationship between EVs and mitochondria in inflammation is an active area of research, although further studies are needed to fully understand the mechanisms involved and how they may be targeted for therapeutic purposes. Here, we have reported and discussed the latest studies focused on this fascinating and recent area of research, discussing of tricky connection between mitochondria and EVs in inflammatory-related diseases.

## 1. Introduction

Mitochondria are intracellular organelles involved in a plethora of functions and activities in the cell, including a tricky role in inter-cellular communication [1,2,3].

Structurally, the mitochondrion is defined by two distinct phospholipidic membranes, the outer mitochondrial membrane (OMM) and the inner mitochondrial membrane (IMM), which delineate two different mitochondrial spaces: the intermembrane space and matrix [4,5]. The OMM allows the passage of water, ions, and small molecules through several channels, such as Voltage-Dependent Anion Channel (VDAC) family, while the IMM is permeable only to water, molecular oxygen (O_2_), and carbon dioxide [6]. This selectivity is fundamental to generate the electrochemical gradient across the membrane required for adenosine triphosphate (ATP) production and for the regulation of ion homeostasis (and signaling), mainly of calcium (Ca^2+^) ions [7,8]. Mitochondrial Ca^2+^ signaling plays a pivotal role in cell fate and for a variety of cell processes, such as bioenergetics, apoptosis, autophagy, and inflammation. In several inflammatory-related pathologies, the mitochondrial Ca^2+^ signal is dysregulated contributing to mitochondrial dysfunctions and at the exacerbation of disease pathogenesis and outcome [9,10,11,12]. Altered mitochondria are implicated in the formation of damage-associated signals and in the activation of the immune system; this suggests that the maintenance of mitochondrial and [Ca^2+^]_m_ homeostasis is a crucial event to avoid the trigger and exacerbation of pathological pathways [13]. Mitochondria may serve as a hub for inflammation, being a source of mitochondrial damage-associated molecular patterns (mtDAMPs), such as mtDNA, radical oxygen species (ROS), and other mitochondrial products [14]. These mitochondrial molecules are freely dumped in the cytosol or in the extracellular milieu to induce the expression and activation of numerous pro-inflammatory cascades and mediators (see below) [15]. In a context of mitochondrial compartmentalization response of inflammation (“mito-inflammation”), the released mitochondrial-derived molecules may act as pro-inflammatory autocrine and paracrine signals to recipient cells, playing a key role in several inflammatory-related diseases. The exchange of mtDAMPs among cells is in part mediated by a subtype of extracellular vesicles (EVs) derived from mitochondria, which drive mitochondrial cargoes, including outer, inner, and matrix content [16,17].

Indeed, cells need to transfer information with each other to coordinate cellular growth and cellular development and to promote their environmental adaptation and function. This intercellular communication may be performed by direct contact between cells or by the release of EVs. The mitochondrial-derived vesicles (MDVs) and mitovesicles are novel subpopulations of EVs of mitochondrial origin; they differ in morphology, size, and content from canonical EV subtypes [18,19]. The levels and cargo of MDVs and mitovesicles may be altered when mitochondrial dysfunction occurs, modulating the metabolic and inflammatory responses of recipient cells [20,21]. The control of MDVs and mitovesicles biogenesis and release might be interesting in view of a possible modulation of inflammation. This could shed a new light in the study of mitochondrial homeostasis and on how its loss, with the consequent release of MDVs and mitovesicles, is linked to the development of various pathologies. In this review, we will discuss the role of EVs in trafficking mitochondrial cargoes during inflammation in several diseases, such as in pulmonary and cardiovascular diseases, neurodegenerative pathologies, and cancer.

## 2. “Mitochondrial Damage-Associated Molecular Patterns” Drive the Mito-Inflammation

Although the original sources of DAMPs comprise also plasma membrane (PM) and nuclear and intracellular proteins, it has been discovered that mitochondrial components, including mtDNA, mitochondrial ROS, ATP, cardiolipin (CL), and Ca^2+^, play a key role to drive inflammation, acting as mtDAMPs (Figure 1) [22,23]. Consistent with the importance of mitochondrial health, several mechanisms, such as MOM permeabilization, mitochondrial permeability transition pore (mPTP) opening, or vesiculation, control the release of mtDAMPs contributing to maintain the mitochondrial quality control. An integrated coordination of mechanisms operates to limit mitochondrial stress by monitoring mitochondrial damage and providing the selective removal of dysfunctional mitochondrial proteins or parts of organelle by mitophagy [24,25].

In order to modulate the immune responses, mtDAMPs may be released accidentally in the cytosol and in the extracellular milieu after unregulated cell death or intentionally mediating mPTP opening and by specific MDV-dependent pathways [18,26,27,28]. The recent literature is increasingly focused on the role of mtDAMPs in influencing the immune system and consequently affecting the progression of inflammatory-related diseases [1,29].

The main mtDAMP that had until now emerged for its contribution to a systemic inflammatory response is mtDNA. mtDNA encodes for a limited number of genes essential for proteins involved in the mitochondrial respiratory chain. Unlike nuclear DNA, mtDNA is hypomethylated and more prone to oxidative damage due to the lack of structural histones and by the proximity to sources of ROS from the electron transport system [30]. This makes mtDNA more prone to damage, contributing to mitochondrial dysfunction and the development of chronic inflammatory-related diseases [21,29].

During stress, mtDNA is released from mitochondria to promote by several mechanisms, including cyclic GMP-AMP (cGAMP) synthase (cGAS) and Stimulator of Interferon response cGAMP Interactor 1 (STING), NLR family pyrin domain containing 3 (NLRP3) and NLR family CARD domain containing 4 (NLRC4) inflammasome, and toll-like receptor 9 (TLR9) signaling, inflammation through the release of pro-inflammatory mediators, and the activation of immune cells [31]. cGAS is a cytosolic protein that responds to released mtDNA catalyzing the formation of cGAMP, which acts as second messenger to STING1 to initiate the inflammatory response through the activation of NF-kB signaling and the synthesis of cytokines, including interleukin-6 (IL-6), Tumor Necrosis Factor alpha (TNF-α), and interferon-β1 [31].

mtDNA is a pivotal inflammasome activator, particularly when oxidated by ROS [32]. The inflammasomes are supramolecular platforms for the activation of caspase, including caspase 1, which contributes to proteolytic maturation of pro-inflammatory cytokines, IL-1β and IL-18 [33]. The inflammasomes, NLRP3 and NLRC4, are direct targets of mtDNA; both interact with procaspase-1 and the adaptor protein ASC (apoptosis-associated speck-like protein containing a caspase recruitment domain). The activation of NLRP3 inflammasome is divided in two phases: priming and activation [34]. In the priming step, the expression and post-translational modifications of different inflammasome components are induced by NF-kB signals in response to pro-inflammatory stimuli. The activation consists of the relocalization of the inflammasome to mitochondria through physical interaction with thioredoxin-interacting protein (TXNIP) and CL in the presence of ROS, resulting in cytokines secretion [35,36]. Mitochondrial membranes have a characteristic lipid composition; in the IMM, the high cardiolipid content contributes up to 20% of total lipids [37,38]. CL is an important phospholipid in the mitochondria; it has an indispensable role in maintaining the electron transport chain [29]. Disturbances in mitochondrial functions lead to potential modifications of CL that may cause its externalization from IMM. Consequently, CL is transferred to the OMM or other cellular structures where it modulates the activation of inflammatory reactions through activation of NLRP3 inflammasome [34].

In the extracellular milieu, mtDNA naked or complexed with the mitochondrial Transcription Factor A (TFAM) protein induces neutrophilic activation in a TLR9-dependent manner, acting as the prototypical TLR9 agonist, the bacterial DNA, promoting same immunostimulatory effects [39].

In addition to mtDNA, the mtROS act as strategic mtDAMPs for inflammasome activation. They are generated as by-product of oxidative phosphorylation at level of complex-I (NADH-CoQ reductase) and complex III (cytochrome c reductase) of ETC, initially generating anion superoxide which is then converted to hydrogen peroxide by mitochondrial superoxide dismutase enzyme [40,41]. The production of mtROS is influenced by mitochondrial membrane potential, metabolic state, and oxygen levels [41]. mtROS promote oxidative damage to mtDNA, mitochondrial proteins, and membranes, altering the functionality and structure of the organelle, producing additional mtROS. Elevated mtROS levels are seen during mitochondrial malfunctioning and may activate redox-sensitive transcription factors, leading to the production of pro-inflammatory cytokines, including IL-1 and IL-8 [42,43,44,45]. The levels of mtROS are also linked to the level of mitochondrial Ca^2+^, which can directly and indirectly stimulate mtROS production: directly, by stimulating mitochondrial resident ROS-generating enzymes, such as ketoglutarate and glycerol 3-phosphate dehydrogenase, or indirectly, by the Ca^2+^-dependent activation of nitric oxide synthase, for which the mediating nitric oxide blocks the mitochondrial complex IV, and/or via reverse electron transport induced by succinate, Nicotinamide adenine dinucleotide concentration, Coenzyme Q pool redox state, and mitochondrial membrane potential [46,47,48]. In turn, mtROS may perturb Ca^2+^ signaling affecting the functionality of Ca^2+^-receptors and Ca^2+^-effectors with consequences in the intracellular signals [49]. An example is represented by oxidative modification of the mitochondrial Ca^2+^ uniporter (MCU) on cysteine 97, which resulted in a persistent channel activation that led to an excess of mitochondrial Ca^2+^ uptake [50]. In general, the perturbation of mitochondrial Ca^2+^ signaling boosts the mtROS production with consequent repercussions on inflammasome activation and the release of proinflammatory mediators [40].

Mitochondria are a major source of ATP; under stress conditions the energy demand increases, leading to rapid changes in ATP levels [29]. High intracellular levels of ATP lead to the extrusion of ATP into the extracellular environment which activates purinergic receptors to trigger inflammatory reactions [51]. Specifically, ATP binds to receptors of the PM family P2X (non-selective cation channels with inotropic ligand): the binding to P2X7 receptors triggers potassium efflux with consequent NLRP3 activation [52]. Within the innate immune system, ATP is capable of triggering chemotaxis and neutrophil adhesion. Indeed, it increases degranulation and increases ROS production to mediate pathogen killing [53,54].

## 3. The New Era of Extracellular Vesicles Communication

In recent years, the intercellular communication exerted by EVs has gained a lot of attention for its potential role in diagnostics and therapeutics [55]. The expression “extracellular vesicles” refers to lipid mono- or bi-layer-delimited particles of distinct composition, content, size, density, and cellular origin [56]. Based on their mode of biogenesis, EVs may be categorized into (I) exosomes (vesicles embedded by the lumen of multivesicular bodies), (II) microvesicles/ectosomes/microparticles (generated via fission or budding of PM), and (III) apoptotic bodies (PM blebbing by cells enduring programmed cell death) (Figure 2) [57,58].

Exosomes refer to small EVs that commonly range between 40 and 150 nm and are formed through the inward budding of endosomal membrane, producing an intraluminal vesicle known as multivesicular body (MVB). The MVBs may either follow the secretory pathway and fuse with the PM to release its content as exosomes for recipient cells or follow the lysosomal pathway where the MVBs fuse with the lysosomes for degradation [58]. Exosomes can be classified into classical exosomes that are characterized by frequently used markers such as CD9, CD63, and CD81 [59] and non-classical exosomes (CD9, CD63, and CD81 negative); both classes present also abundant proteins including Syntenin-1 and ALIX [60].

Microvesicles, also termed ectosomes or microparticles, have a diameter ranging from 100 to 1000 nm [61]. They are formed through outward budding of the PM, triggered by molecular rearrangements in each of the two leaflets of PM bilayer [62]. Since microvesicles originate from the PM, they contain cytoplasm-residing components, such as enzymes, mitochondrial, ribosomal, and centrosome proteins. In contrast to exosomes and apoptotic bodies, the microvesicles express membrane-associated proteins annexin A1 and A2, which are considered specific markers of non-exosomal small to large EVs [60]. Belonging to this class of EVs, apoptotic bodies, with a range from 100 nm to 5000 nm, are formed by a cell shrinkage which lead to cell death by drifting them from the membrane [63,64].

Although the classification of EVs based on biogenesis has improved the understanding of roles of EVs, it does not consider that EVs from different origins may have similar densities and sizes or that vesicles belonging to the same class of EVs may express different markers and carry diverse cargoes. Therefore, while prevalent themes have emerged, there is considerable heterogeneity in the structures and functions described for EVs [65]. In fact, the International Society for Extracellular Vesicles has recommended the non-specific term “EVs” for the vesicles released from cells [56]. Amounts of substances including proteins, lipids, DNA and organelles are contained within EVs, which can be released between cells with specific advantages, such as (I) protection, made by the membrane structure of EVs that can protect the cargo from degradation, (II) targeting by markers localized on the surface of EVs that can bind to the corresponding receptors on the recipient cells to achieve specific targeting, and (III) enrichment of EV contents that are discharged around or inside the recipient cells [58,66,67,68]. These characteristics underline the importance of EVs in the intercellular communication and highlight their potential role as markers, therapeutic targets, and drug carriers [69,70,71]. Recent evidence has revealed that EVs also contain material from cellular compartment, including mitochondria. The detection of mitochondrial content (mtDNA and proteins) enclosed by EVs has led to the recognition of different biological functions of mitochondrial cargoes, such as inflammasome activation, mitochondrial degradation, and metabolic modulation [27]. In this context, a novel study reported that the incorporation of mitochondria from EVs derived from neural stem cells (NSC) into inflammatory mononuclear phagocytes reestablished physiological mitochondrial dynamics and cellular metabolism, diminishing the expression of pro-inflammatory markers in target cells [72].

Mitochondria can provide their own vesicles called mitochondrial-derived vesicles (MDVs), which were first described in 2008 as vesicles released by mitochondria carrying selective cargoes [73]. MDV production has been mainly linked to elevated levels of intracellular ROS [74], suggesting that it is part of the quality control process complementary to mitophagy. Indeed, a PINK/Parkin-regulated process is characterized by TOM20-positive MDVs that contain OMM proteins. Under oxidative stress, the enveloping of oxidized mitochondrial components into TOM20-positive MDVs targeted to Tollip (Toll-interacting protein)-positive endosomes facilitates MDV trafficking to the lysosomes [75]. Another secretory signaling pathway involves the fusion of MDVs with MVBs, where IMM/matrix incorporation to EVs depends on OPA1 and Snx9, in contrast to the formation of TOM20-positive MDVs [20,74]. In 2021, a new subgroup of mitochondrial EVs was identified by D’Acunzo et al., with the term “mitovesicles”, indicating a double-membraned EVs carrying mitochondrial proteins [19]. The researchers have used a high-resolution gradient separation to isolate brain-derived EVs from WT mice to obtain eight fractions containing different-sized EVs carrying various contents. The fraction with highest density did not reveal the regularly used markers TSG101, Alix, and CD63 but was enriched in mitochondrial proteins such as VDAC, COX-IV, and PDH-E1. The “mitovesicles” are also characterized by an elevated content of CL, as well as by enzymatic activity and ability to produce ATP. However, a clear classification of “mitovesicles” and the differences related to MDVs require further investigation. The differences among these subgroups regarding mitochondrial cargoes and expression of different markers also need to be explored in other physio-pathological conditions.

Other studies revealed the presence of intact mitochondria within EVs in human brain endothelial cells (BECs) that can be transferred from the parent/donor to recipient cells, specifically during stress and injury [76]. Interestingly, transferred mitochondria localized with the mitochondrial network of recipient cells, resulting in increased cellular ATP production [77,78]. Indeed, recent evidence highlighted that intact mitochondria within EVs, released from NSC, conserved mitochondrial membrane potential and respiration. Furthermore, the incorporation of NSC-derived mitochondria in inflammatory mononuclear phagocytes, mediating EVs, restored normal mitochondrial dynamics and cellular metabolism reducing the expression of pro-inflammatory markers in the phagocyte cells [72]. This evidence provides the indication that functional mitochondria may be exchanged, via EVs, to recipient cells to influence the mitochondrial homeostasis, opening a new way for the evolution of novel cellular approaches aimed at restoring mitochondrial dysfunction.

## 4. The Contribution of Mitochondria and EVs in Inflammatory-Related Diseases

This chapter will focus attention on the increasingly debated connection among mitochondria, EVs, and inflammation. The signaling of mitochondria and EV will be discussed considering various human districts to better clarify their roles in the inflammatory-related diseases (Figure 3).

### 4.1. Pulmonary System

The pulmonary system is the principal organ that provides defense against viral and bacterial pathogens in the lungs. Communication between structural and immune cells is indispensable for stimulating immune mechanisms and maintaining homeostasis. In recent years, the intercellular communication in the lungs mediated by EVs has gained much attention as reviewed by Bartel and colleagues [79].

EVs can be produced and released by all cell types of the pulmonary system. An increasing number of studies have characterized EVs from bronchial epithelial cells (BECs), alveolar macrophages, fibroblasts, type II pneumocytes, and vascular endothelial cells under physiological or pathological conditions [80]. Environmental components, such as cigarette smoking (CS), pollutants, and microbial pathogens that induce injury in these cells, contribute to EV release [81]. Specifically, it has been demonstrated that EV contents provide biological information about the physiological/pathological conditions of the lungs [57]. Accordingly, EVs act as mediators in the airway microenvironment and play a crucial role in the production of various pro-inflammatory stimuli, serving as important elements in the lung inflammatory process. As a sensor of inflammation, during environmental injury and infections, mitochondria may respond to these stimuli through altered mitochondrial dynamics and dysfunction [82,83]. Consequently, damaged mitochondria impact Ca^2+^ homeostasis and metabolism, with repercussions on proliferation, apoptosis, airway contractility, responses to oxidative stress, and fibrosis, which are all hallmarks of airway disease pathophysiology [82,83,84,85].

As mentioned above, mtDAMPs are the main mitochondrial damage-associated signals released under stress conditions. Among these, mtROS represent one of the excessive mtDAMPs in Cystic Fibrosis (CF) airway cells [15]. CF is an autosomal recessive condition that affects various organs and tissues, characterized by mutations of the Cystic Fibrosis Transmembrane conductance Regulator (CFTR) channel, a protein that regulates the exchange of chloride, bicarbonate, and sodium ions through epithelial membranes [86]. In CF, loss or malfunction of CFTR causes alteration in ion flux resulting in the production of viscous mucus and, consequently, in obstruction of small- and medium-size bronchioles and bronchiectasis [86]. This leads to the promotion of recurrent bacterial infections and thus inflammation, which may proceed until patients die from respiratory insufficiency [87]. The persistent bacterial infections, particularly of *P. aeruginosa*, promote mitochondrial Ca^2+^-overload and excessive mtROS production in human CF bronchial cells, which in turn induce the activation of NLRP3 and NLRC4 inflammasomes and subsequent release of IL-1β and IL-18 [88]. The high levels of mtROS lead to promote other mitochondrial impairments in a feedback stimulatory manner that sustains the activation of oxidative-sensitive transcription factors, including NF-kB, HIF1, and AP-1, which exacerbate the chronic pulmonary inflammation favoring an elevated production of cytokines/chemokines and priming of the inflammasome NLRP3 and its members [40,89,90,91,92,93]. This inflammatory state that characterizes CF airway cells could be mediated by EVs, that seem to have an increasingly important role in the pathogenesis of chronic respiratory diseases. In fact, a study reported a higher EVs concentration in CF bronchial supernatants and in bronchoalveolar lavage fluid (BALF) from CF patients compared to controls. Interestingly, EV release increased with age in CF patients, suggesting a potential role of EVs in the progression of disease. On the contrary, the increased expression and stability of CFTR after administration of AKT inhibitors and CFTR correctors was responsible for the reduction of EV release [94,95]. The reduction of EV release is also correlated with an increase in autophagy induction (LC3-II) in CF airway cells following treatment with AKT inhibitors and CFTR correctors. This consideration could be interpreted by a coordinated relationship between the EVs release and autophagy pathways for the maintenance of cellular fitness [96]. The inability of the F508delCFTR protein to produce an exact folded state disrupts cellular networks which preserve the cell from acute stress, and exosome secretion could provide a different way to relieve stress when recycling pathways are compromised [97]. Moreover, exosomes derived from CF epithelial cells mediated inflammation by controlling the migration and the activation of neutrophilic leukocytes in the CF airways. This is supported by the rich content of the CF airway bronchial-epithelial-cell (CFBE41o-)-derived exosomes in the integrin proteins, such as VCAM1, with resultant movement of neutrophils to inflammatory sites, and in the ligand S100 A12, which can bind the RAGE receptor sited on the surface of the receiving neutrophils, responsible for their activation [98,99]. Recently, Forrest et al. demonstrated that CF sputum-derived EVs may activate naïve neutrophils, inducing both the exocytosis of their primary granule and their concomitant caspase-1 activation and IL-1β production [100]. Furthermore, EVs released by activated neutrophils favored the transfer of active caspase-1 to primary tracheal epithelial cells which, in turn, stimulate the inflammasome and release of IL-1α, IL-1β, and IL-18.

As evidenced, there is an interesting potential for the application of EVs, microvesicles, and exosomes as vehicles for delivering exogenous CFTR to CF cells [101]. This study revealed that EVs, microvesicles, and exosomes could be used to deliver exogenous CFTR glycoprotein and its encoding mRNA (mRNA (GFP-CFTR)) to CF cells, correcting the CFTR chloride channel function. Collectively, the study suggests the potential use of microvesicles and exosomes as vectors for transferring and functionally correcting the genetic defect of CFTR in human CF cells [101]. Additionally, examining exosomes derived from CF patients could be helpful in characterizing the pathogenesis of the disease.

Many studies have demonstrated that EVs can be used as an important biomarker in Chronic Obstructive Pulmonary Disease (COPD) diagnosis and prognosis. COPD, mainly caused by cigarette smoke (CS), is characterized by small-airway inflammation, narrowing, and emphysema, which causes irreversible lung damage [102,103]. Inflammation and tissue remodeling through the modification of bronchial epithelial cells, differentiation of fibroblasts into myofibroblasts, and eventually the EVs produced by these cells represent the main actors involved in this disease [104,105]. Takahashi et al. showed that endothelial cell-derived EVs were abundant in COPD patients, especially during exacerbations, highlighting the potential of EVs to predict these events [106]. Moreover, studies revealed that some miRNAs found within EVs were directly associated with COPD severity due to their involvement in disease development and progression [107]. CS exposure may modify the cargo of EVs, released by BECs, with upregulation of several miRNAs (miR-500a-5p, miR-574-5p, miR-656-5p, miR-3180-5p, and miR-3913-5p) and downregulation of three miRNAs (miR-222-5p, miR-618, and miR-130b-5p) [108]. Furthermore, in another study, the authors reported that microparticles can promote the release of a subset of miRNAs (miR-125a, miR-126, and miR-191) to resident macrophages, which are responsible for the removal of apoptotic cells, highlighting the effects of EVs transfer on COPD pathogenesis [109].

Recently, several studies have focused on the role of EVs in SARS-CoV-2 infection [110]. It has been demonstrated that different cytotypes represent the first source of EVs including epithelial cells, macrophages, neutrophils, and endothelial cells [111,112,113]. The presence of ACE2 (receptor angiotensin-converting enzyme 2) in EVs produced by endothelial progenitor cells suggests a crucial role of these vesicles in promoting infection, given that ACE2 is fundamental for the fusion between SARS-CoV-2 viral particles and the host cell membrane [114]. During respiratory infections by coronaviruses, the level of EVs in the serum was augmented [115,116]. A similar effect was emerged also with exosomes, where a systemic increase in exosomes has been detected during SARS-CoV-2 infection [117,118]. Moreover, it has been demonstrated that EVs may activate host immune responses when their transfer viral and self-antigens [119].

In recent years, there has been increased attention on inflammatory-related diseases due to the “paracrine ability” of mesenchymal stem cells (MSCs), which can release immunomodulatory cytokines and tissue repair growth factors [120]. Evidence shows that MSCs are involved in mitochondrial transfer in the respiratory tract, as MSC-derived EVs contain mitochondria [76,77,120,121,122]. The release of respiration-competent mitochondria through microvesicles has been extensively investigated in MSCs, as they can help recover cells from insult [76]. Spees et al. demonstrated that mitochondrial transfer from MSCs may reestablish aerobic respiration in mitochondria-depleted A549ρ° cells, and the delivery of MSC-derived mitochondria to pulmonary epithelial cells has been associated with reduced inflammation in vivo models of acute lung injury and asthma [76,77,123,124]. Furthermore, EV-mediated mitochondrial transfer is essential for MSC to repair human distal lung epithelial cells [122]. In fact, a recent study revealed that LPS administration promoted an important impairment of barrier integrity of primary human small airway epithelial and microvascular endothelial cells, associated with massive mitochondrial dysfunction and alterations in mitochondrial biogenesis and mitophagy. In an LPS-induced lung injury model of acute respiratory distress syndrome (ARDS), for a hyperinflammatory lung disease characterized by a damaged alveolar epithelial–endothelial barrier, the administration of MSC-derived EVs reduced the LPS-stimulated increases in BALF total protein associated with LPS-dependent tissue injury and, at the same time, diminished the neutrophils recruitment, favoring the recovery of mitochondrial respiration and ATP production in lung tissue [125]. The lungs were subsequently recovered after 24 h (LPS instillation) in an LPS-induced acute lung injury (ALI) model, where the survival of mice increased with MSC instillation [77]. In a different model of acute lung injury induced by *Escherichia coli* LPS, the mouse bone marrow-derived stromal cells were able to transfer mitochondria to alveolar epithelium increasing alveolar ATP levels [126]. Moreover, induced pluripotent stem cells (iPSC-MSCs) not only delivered mitochondria to damaged bronchial epithelial cells but also transferred mitochondria to airway smooth cells, alleviating the inflammation in human lung cells and in murine lungs exposed to CS [127,128].

### 4.2. Cardiovascular System

In the etiology of most cardiovascular diseases (CVDs), the inflammatory process is considered a crucial checkpoint in the progression of the pathology. Growing evidence on the mechanisms behind inflammation in the vascular endothelium implicates the existence of EVs as crucial molecular mediators between donor and recipient cells [129]. Although inflammation is also orchestrated by mitochondria and their dysfunctions have long been considered a hallmark of CVDs, there is still a lack of evidence linking EV cargo and mitochondria within the inflammatory response of the heart [130,131].

To the best of our knowledge, a role for mitochondrial components carried by EVs at the cardiomyocyte level has been reported to be involved in mitochondrial quality control mechanisms and bioenergetics status in ischemic heart disease [132]. Recently, a research group identified cardiomyocytes-released EVs with a size range of 220–600 nm by electron and confocal microscopy techniques. These EVs were able to incorporate mitochondrial proteins from donor cells, belonging to the electron transport chain (ETC), as well as ATP, mRNA, and mtDNA. Fusion of EVs with target cardiomyocytes exposed to hypoxia and reoxygenation improved their bioenergetics and mitochondrial biogenesis by directing the cargo in a very fast and efficient manner: 3–4 h for the complete fusion and delivery, with an absence of mitochondrial degradation as seen in naked mitochondrial transfer [132].

However, either physiologic or pathologic response of recipient cells is heterogenous and depends on the content of EVs that are going to be transferred. Intriguingly, EVs carrying mtDAMPs can have proinflammatory roles on the vascular endothelium [133]. LPS-activated monocytes released EVs enriched with TOM22 protein and mitochondria-associated TNF-α and mRNA; once in the endothelium they prompted TNF-α and interferon (IFN) I response contributing to the inflammatory cascade [133]. Accordingly, the same findings have been confirmed in vivo by analyzing circulating proteins in the serum of patients with an active state of inflammation. Of note, EVs from the same monocytes, which were pretreated with antioxidants and with a limited mitochondrial respiration, failed to induce the described proinflammatory behavior.

Monocytic EVs are also responsible for the delivery of other mitochondrial constituents in the vascular endothelium, which are responsible for its damage. This is the case of the mature form of IL-1β accompanied by other organelles fractions enriched in NLRP3, ASC, and cleaved caspase-1 proteins [134]. All constituents of an active inflammasome that is being assembled and packaged in EVs further sustain the inflammatory response once delivered.

Inflammation also characterizes the maladaptive stages of cardiac hypertrophy with an inflammatory signature depicted by the presence of IL-1β, IL-6, and TNF-α [135]. Recent research detected a robust release of EVs carrying parts of mitochondria with proinflammatory functions [136]. Specifically, EV-embedded mitochondria had a decreased expression of scavenger enzymes such as superoxide dismutase 1 (SOD1), VDAC, and mitochondrial complexes. Once transferred to donor cells, they were able to induce inflammation and oxidative stress by altering IL-1β and TNF-α levels and by inducing the expression of Inter-Cellular Adhesion Molecule 1 (ICAM-1) and Vascular Cell Adhesion Molecule 1 (VCAM-1) [136]. MAPK-Activated Protein Kinase 2 (MK2) is at the middle of this EV-mediated process of dysfunctional mitochondria transfer to target cells; its silencing prevented the cargo export in charge of endothelial cells and the inflammatory-related contribution to cardiac hypertrophy [136].

One of the most well-characterized inflammatory pathways is associated with atherosclerosis [137]. Atherogenesis includes at least five pathological steps divided into endothelial dysfunction, formation of fatty streak, migration of leukocytes and muscle cells into the wall vessel, foam cell formation, and degradation of the extracellular matrix. Updated techniques in EV isolation allowed researchers to detect their presence in the lesion site, understand the source of the release, and evaluate their contribution in all pathological steps [138]. Despite this effort, caution should be exercised when interpreting related mechanistic insights, as they were obtained from in vitro studies and may differ significantly from what happens in vivo. Major findings reported EVs as released from macrophages, lymphocytes, and granulocytes, but a modest amount has been detected from erythrocytes and smooth cells in the plaque [138]. The communication established by EVs is not one way; receiving cells, in turn, can send EV-mediated messages to the donor, as evidenced in endothelial cells in response to leukocyte-induced ROS and inflammation [139]. Of note, there are no EVs released in the plaque from platelets. Platelet-derived EVs were only detectable in the blood.

During atherosclerosis, EVs exhibit pro-thrombotic functions and are actively involved in all stages of the disease [140]. In the early phases, they regulate the adhesion of leukocytes to the endothelial wall, taking advantage from the transfer of ICAM-1 via monocytes’ EVs and through the activation of the CCL2-dependent NF-kB pathway [141]. The same route has been shown to be enhanced by miR-155 delivery from neutrophils [142]. Although VCAM-1 and Selectins are also essential for this process, they are not involved in this delivery system composed of monocytes and neutrophiles. Interestingly, evidence supports a role for adipocytes-mediated release of EVs in the increased expression of VCAM-1 after being exposed to hypoxia and TNF-α, elements mimicking obesity conditions, and P-selectin and Glycoprotein Ib platelet subunit alpha (GPIbα) from circulating platelets directed to monocytes and neutrophiles, respectively [143,144,145].

Once monocytes have migrated into the endothelial wall, they differentiate into macrophages. Their polarization and further conversion into foam cells are crucial events in the intermediate stages of atherosclerosis. The EVs released from platelets enhance this process by promoting phagocytosis of oxidized LDL, while adipose-derived EVs mediate the polarization of macrophages into the M1 state [146,147]. The opposite effect (M2 macrophage polarization), although not well-characterized, appears to be mediated by EV activity of endothelial cells exposed to oxidized LDL and oxidative stress [148].

Plaque maturation includes calcification, with drastic consequences in the outcome of patients affected; also, calcification has an inflammatory-based origin [149,150]. The analysis of carotid plaques from 136 patients highlighted the presence of EVs released from inflammatory macrophages; the cargo was mainly composed by S100A9 and Annexin V which mediated the process of calcification together to an increased expression of the receptor for advanced glycation end products (RAGE) conferring to EVs the significant alkaline phosphatase activity [151]. Once again, the great biological potential of EVs as mediators of inflammation resides in both their cargo and donor cells entity. It has been demonstrated that the silencing of S100A9 in the donor cells (macrophages) or the same experiments performed in a mouse model of genetic ablation of that protein decreased the inflammatory potential and the calcification of plaques [151].

### 4.3. Nervous System

The presence of the inflammatory status is a common feature of several neurodegenerative conditions and represents the primary cause of neuronal dysfunctions and loss of the brain cell population [152]. However, a complete understanding of the pathological mechanisms mediating the onset and the progression of inflammation in the brain is still lacking. Interestingly, there is a growing recognition that proper functioning of the mitochondrial compartment is closely related to inflammation. Indeed, loss of mitochondrial homeostasis has been found in almost all known neurodegenerative condition and the release of mtDAMPs in the cytosol and in the extracellular environment via an EV-dependent manner is a phenomenon commonly observed in neurodegeneration [153]. The principal mtDAMP associated with inflammation in several neurodegenerative conditions, including Huntington’s disease (HD), amyotrophic lateral sclerosis, Parkinson’s Disease, and Alzheimer’s Disease (AD), is the mtDNA [154,155]. Additionally, the release of cytochrome C (cyt-c), which is commonly used as a marker of apoptosis both in vivo and in vitro, has been found to work as an mtDAMP, showing that extra-mitochondrial cyt-c led to chronic inflammation in neuronal culture, increasing the secretion of IL-1β and IL-8 [156,157,158]. Similar observations have been found for the IMM phospholipid CL, which following a mitochondrial damage was released from the organelle to activate the NLRP3 inflammasome, to regulate the phagocytosis and the levels of pro-inflammatory cytokines of adjacent immune cells in the Central Nervous System (CNS) [36,159].

Interestingly, recent insights unveiled that mtDAMPs are highly enriched inside EVs, propagating neuroinflammation and therefore contributing to neurodegenerative conditions [160]. Consistently, fibroblasts and neural stem cells (NSC) obtained from patients affected by HD displayed a high release of EVs transporting increased levels of mitochondrial genetic material [161]. Increased EV and mtDNA levels were also detected in human plasma of HD patients, and their levels changed between premanifest and manifest patients [161]. A study aimed at detecting mitochondrial contents in circulating EVs in the serum of PD patients has been performed to demonstrate the possibility of associating the rate of mtDNA (and other mitochondrial elements) in EVs with the course of PD neurodegenerative conditions, confirming the existence of a mitochondrial signature circulating in EVs associated with a specific inflammatory profile in PD patients [162,163]. In PD, the inflammation is sustained by the occurring mitochondrial dysfunction, which leads to an increase of mtDAMP release, particularly of mtROS, although the levels of circulating mtDNA in the resulting cerebrospinal fluid was reduced in PD patients compared to controls [164]. The reduced levels of circulating mtDNA were justified by I) diminished TFAM and TFB2M transcriptional activity, II) the presence of treatments, and III) impaired mitochondrial biogenesis [154,165], whereas the increased mtROS production and Ca^2+^ imbalance, due to impaired ETC function, contributed to exacerbating the inflammation and PD pathology, affecting mitophagy, which failed in removing damaged mitochondria [166]. Defects in mitophagy and mutations in PINK1 and Parkin are also responsible for aberrant mitochondrial antigen presentation via MDVs, which triggers autoimmune mechanisms in PD [167]. PINK1 and Parkin wild type, under stress, regulate the lysosomal degradation of damaged mitochondrial components via MDV formation, suppressing the mitochondrial antigen presentation mediating the proteasomal turnover of Sorting nexin-9 [167,168].

Recently, the transplantation of NSC has been described as a reliable method to ameliorate the clinical score of animal models of neurodegeneration, and it has been found that EVs may promote the exchange of therapeutic cargoes from NSCs to the host cells [169]. Multiple Sclerosis (MS) is a chronic multifactorial inflammatory demyelinating disease of CNS [170]. Although the etiology of MS is still unclear, it has been demonstrated that mitochondria play an important role [171]. Excessive mtROS production, mtDNA mutation, decreased mitochondrial ATP production, and impairment in mechanisms aimed at preserving of healthy mitochondrial population have also been associated with the onset and progression of MS [172,173]. Intriguingly, a recent work displayed that exogenous NSC transplanted into an animal model of MS (the chronic experimental autoimmune encephalomyelitis model) transfers functional mitochondria to host cells to reduce the neuroinflammation and, in turn, decrease clinical deficits [72]. This work, analyzing which cellular population mainly received the new pool of mitochondria, also found that many of the transferred mitochondria were directed towards mononuclear phagocytes and astrocytes. Additionally, functional mitochondrial constituents have been observed in neuronally enriched EVs isolated from the plasma of an MS patient [174]. This study aimed to analyze the mitochondrial complex activity in neuronal EVs to predict the levels of neurodegeneration in MS. Higher mitochondrial complex IV activity and lower mitochondrial complex V activity were associated with faster whole brain, brain substructure, and retinal atrophy [174].

A recent investigation found increased EV levels in postmortem blood and brain samples from Alzheimer’s disease (AD) patients. In detail, the authors found that sporadic AD patients presented an increase in endothelial- and leukocyte-derived EVs enriched with mitochondrial markers while platelet-derived EVs were found in familial AD patients. Interestingly, these AD-associated vesicles, with different origins, were characterized by an augmented presence of mitochondrial markers and inflammatory mediators and were found to be sufficient to activate neuroinflammation in human cortical organoids [175]. AD is a neurodegenerative disease caused by an excessive accumulation of amyloid β (Aβ) plaques and tau-containing neurofibrillary tangles which result in neuroinflammation and neuronal loss. A large body of evidence suggests that the presence of a damaged population of mitochondria covers an important role in the pathogenesis of the disease. Consistently, oxidative stress, dysfunctional mitochondria, and disrupted mechanism controlling the removal of damaged mitochondria are present in tissues samples of AD patients [176,177,178]. The presence of mitochondrial contents, such as proteins and RNA, and portions of mitochondrial structures have been also found in isolated EVs from astrocytes, neurons, and microglia exposed to Aβ and oxidative stress stimuli, suggesting that part of the mitochondrial population in AD is extruded following a vesicle pathway in the different cell types [179]. Moreover, increased levels of mitochondrial RNA were found in EVs purified from the plasma of AD affected patients and from individuals with mild cognitive impairment, a type of dementia often related to AD [179]. Another study showed higher levels of mitochondrial proteins, such as TOM20 and OPA1, enriched in EVs isolated from astrocytoma exposed to toxic Aβ with respect to non-treated cells [179]. Overall, these data suggest that analyzing the presence of mitochondrial elements in circulating EVs obtained from AD-affected patients could represent a promising method to fill an important gap related to this disease and to several CNS disorders, which is lack of biomarkers for early diagnosis. In line with this, the levels of mtDNA markers contained in EVs isolated from children with autism spectrum disorder were found to be higher than those in normotypic controls and stimulated the release of pro-inflammatory cytokines in cultured human microglia [180]. Elevated levels of mtDNA and mitochondrial proteins have been observed in brain-derived “mitovesicles” isolated from Down syndrome mice compared to controls [19]. This study showed that also the “mitovesicles”, similar to MDVs or EVs enriched with mitochondrial contents, may serve as a biomarker to evaluate the status of mitochondrial dysfunction in neurological diseases. Mitovesicles differ from EVs and MDVs in structure and cargoes, as described before. They are vesicles with enzymatic activity and the capacity for ATP production, but the reduced amount of Ubiquinol-Cytochrome C Reductase Core Protein 2 (UQCRC2) and Succinate dehydrogenase iron-sulfur subunit (SDH-B) in mitovesicles isolated from a Down syndrome brain suggests an impairment of ATP production in brains derived by trisomic mice [19].

Finally, the mitochondrial transfer capacity of EVs has been utilized to counteract the progression of Leber’s hereditary optic neuropathy (LHON), one of the most common mitochondrial diseases [181]. It has been suggested that by using EVs carrying mitochondria, it could be possible to replace the corrupted mitochondria population present in a LHON-affected individual with a healthy and functional pool of mitochondria [182].

All these data show the complex role of the several mitochondria-specific vesicles in neuronal disorders, and ambivalent mitochondrial cargo effects are emerging: I) attenuating the inflammatory response through the extrusion of damaged mitochondria from cells with engulfed mitophagy, using the mitochondrial-specific vesicles-dependent manner, and II) enhancing the inflammatory response through the presentation of mitochondrial markers mediating mitochondrial-specific vesicles and their internalization in immune recipient cells.

### 4.4. Cancer

Cancer cells undergo a variety of metabolic shifts that enable them to support increased proliferation and survive environmental stressors. Cancer has different metabolic adaptation strategies, but many types of cancer are linked to a reprogramming of mitochondria [183]. The Warburg effect consists of a significant increase in the glycolytic rate independent of O_2_ concentration, and it represents one of the common alterations. Although glycolysis is prevalent for energy production, mitochondrial oxidative phosphorylation (OXPHOS) is still necessary to support tumor formation, progression, and metastasis [184,185].

Remarkably, EVs have been found to regulate glucose metabolism in the context of cancer through a multitude of cargo molecules, such as long non-coding RNA (lncRNA), miRNA, and metabolites [186]. A recent study has broadened the knowledge of this expanding field by showing that lncRNAs packaged in EVs can shuttle from inflammatory cells to alter tumor metabolism [187]. Additionally, EVs released from primary tumors may circulate to remote organs to establish a premetastatic niche [188,189].

Furthermore, fatty acid metabolism, which occurs only within the mitochondrial matrix, is also impaired in cancer. Even in this case, EVs are implicated in lipid metabolism [190]. Intriguingly, researchers have shown that via exosomes, cancer-related fibroblasts transport a variety of metabolites to cancer cells to supply substances for the TCA cycle, promoting cancer progression [191]. In addition, adipocyte-derived EVs transported proteins and fatty acid substrates to stimulate fatty acid oxidation in melanoma cells, modifying mitochondrial dynamics and, subsequently, promoting melanoma migration [192].

Inflammation is also known to play a complex role in cancer development. While inflammation can have tumor-inhibitory effects by recruiting immune cells that target tumor cells, it may also have oncogenic effects by supplying growth factors that promote proliferation, invasion, and metastasis [193]. Emerging evidence suggests that inflammatory mediators, such as STING, may play a role in cancer pathogenesis. For example, STING-deficient mice are highly resistant to chemically induced skin cancer, possibly due to reduced inflammatory cytokine production [194]. However, loss of STING can also promote colitis-associated cancer by enabling microbes to colonize damaged gut areas and trigger chronic inflammation [195].

The deregulation of inflammatory responses elicited by mitochondrial dysfunction has broad and context-dependent effects on cancer pathogenesis. The inhibition of MOM permeabilization or MOM permeabilization-driven inflammatory responses reflects, for example, in the hyperactivation of autophagy in cancer cells and promotes cancer progression by compromising immunosurveillance [196,197]. Intriguingly, several tumor cell types exploit autophagy and MOM permeabilization-driven activation of caspase 3 to avoid anti-tumor immune responses elicited by radiotherapy through mtDNA-dependent cGAS signaling [198,199,200]. The new findings raise the possibility that, in BAK1-competent tumor cells, pharmacological BCL-2- associated X (BAX, apoptosis regulator) inhibitors may accelerate MOM permeabilization-driven mtDNA release, inducing cGAS signaling prior to caspase-dependent cleavage and inactivation of cGAS [201].

Studies have shown that ROS-mediated damage leads to various genomic alterations, including mutations, deletions, and chromosomal translocations. Some studies suggest that the maintenance of mitochondrial genome integrity may play a role in regulating oncogenesis. For example, cytoplasts (cells lacking nuclear DNA) that bear an mtDNA mutation in the gene encoding ND6 (a complex I subunit) have been fused to cells with low tumorigenicity, resulting in high ROS induction, increased tumor outgrowth, and apoptotic resistance [202].

Damaged mtDNA plays a complex role in cancer development and progression, with effects ranging from promoting tumor growth and metastasis to inducing pro-inflammatory and anti-inflammatory responses in the tumor microenvironment (TME). Mitochondrial dysfunction caused by mtDNA mutations has also been shown to promote the cell remodeling of cancer cells. The acquisition of mtDNA from host cells partially restores mitochondrial function in tumor cells, re-establishing respiration to promote tumor growth. Lung metastatic tumor cells have exhibited complete restoration of respiratory function, indicating that mtDNA transfer from host cells in the TME to tumor cells with impaired respiratory function overcomes the pathophysiological process of mtDNA damage and supports the high plasticity of tumor cells [203]. Interestingly, damaged mtDNA is released into the cytoplasm through mitochondrial membrane channels, and extracellular mtDNA has been shown to promote both pro-inflammatory and anti-inflammatory effects in the TME. Activated neutrophils can expel extracellular traps (NETs) containing mtDNA, which promote the release of interferons and other pro-inflammatory cytokines to form a chronic inflammatory environment that contributes to cancer cell growth and metastasis [204,205,206,207,208]. Conversely, recognizing the damaged mtDNA, dendritic cells up-regulate CD86, CD83, and human leukocyte antigen (HLA-DQ) expression and enhance the transcription and release of TNF-α and other inflammatory factors, thereby contributing to forming an anti-tumor inflammatory microenvironment [209,210].

Furthermore, although not related to inflammatory responses, the horizontal transfer of mtDNA contained in MDVs from cancer-associated fibroblasts to cancer cells has been noted in several models of hormone-resistant breast cancer [211]. These studies exemplify scenarios in which mtDNA can be released (and possibly function as a mtDAMP) by cells that are not failing to regulate cell death and suggest the tricky connection between mtDAMPs and EVs in a TME context.

Tumors are intricate systems constantly in communication with their microenvironment, upon which they depend for growth and survival. EVs actively contribute to remodeling the TME and priming metastatic niches to support tumor progression and survival [212]. The TME is composed of immune cells, vessels, tissue stromal cells, adipocytes, extracellular matrix proteins, and soluble factors. The TME is crucial for supporting tumor cells at all stages of their development and is modulated by intercellular communication [213]. EVs are able to regulate functional interactions between tumor cells and non-malignant cells in the TME.

To the best of our knowledge, tumor cells may regulate the immune response by secreting tumor-derived exosomes (TEXs) that contain tumor antigens and immune-stimulatory or immune-suppressive signaling molecules. The study of TEXs is an emerging field in cancer research, with the potential to provide new diagnostic and therapeutic strategies for cancer treatment [214]. TEXs can be taken up by dendritic cells and processed into tumor antigen-loaded dendritic cells, which can increase the number of CD8+ cytotoxic T-lymphocytes. However, the direct activation of T cells by cancer exosomes has not been reported, and the T-cell stimulation function of cancer exosomes requires uptake and processing of tumor antigens by dendritic cells. Intriguingly, recent evidence showed that circulating exosomes can be used as biological markers for pancreatic cancers [189].

## 5. Conclusions

In the last few decades, the field of EVs has rapidly expanded revealing the crucial involvement of EVs in a plethora of physiological and pathological processes, including cancer, neurodegeneration, and cardiovascular and pulmonary diseases. It is now widely recognized that EV production is not restricted to a specific cell type but occurs in vitro and in vivo among different cellular subsets, thus increasing the interest of scientific community to this compelling form of intercellular correspondence.

The identification of EVs of mitochondrial origin has paved the way for the understanding of new possible biological functions of the EVs. Within this class of EVs, mitovesicles and MDVs can be distinguished by their morphology and content.

Despite mitochondria-originated vesicles being initially thought of solely as a quality control mechanism by which cells promote the ejection of damaged or dysfunctional portions of mitochondria into the extracellular space, the discovery of the incorporation of MDVs and mitovesicles into the mitochondrial network in recipient cells suggests that there is a more complex significance than just a role in cell protection. These subclasses of EVs are gaining great interest especially in the inflammatory field because, under pathological or mitochondrial stress conditions, cells release MDVs and mitovesicles to transfer mtDAMPs or mt-proteins to recipient cells. The exchange of this mitochondrial material may trigger either anti- or pro-inflammatory responses in receiving cells, representing an important tool of cell-to-cell communication indispensable for controlling complex biological processes, such as inflammation.

Although further studies are needed to understand the molecular mechanisms driving MDV and mitovesicle formation and release, a plausible candidate involved in this signaling pathway could be represented by Ca^2+^. Indeed, it has been widely reported that this bivalent cation is intrinsically linked to mitochondrial dysfunction and bioenergetics [9]. Particularly, upon stress stimuli, mitochondrial Ca^2+^ overload promotes a series of events crucial to induce oxidative stress and inflammation through the production of mtROS and the release of mtDAMPs. Given the tight association between mitochondrial stress and the release of MDVs and mitovesicles, it would not be surprising that Ca^2+^ might play a key role also in controlling MDV and mitovesicle biogenesis, release, and cargo embedding.

Thus, the advances made by biotechnology over the years have made it possible to distinguish different subtypes of EVs that differ in origin, composition, content, size, and density. Although the categorization of this large class of small vesicles has been useful in improving understanding of the role of EVs in some contexts, several open questions need to be answered. Little it is known about their roles in mitochondrial regulation of recipient cells, what signals drive the formation and release of a particular type of EV from a donor cell, and how EVs recognize a specific recipient cell.

## Figures and Tables

**Figure 1 ijms-24-08181-f001:**
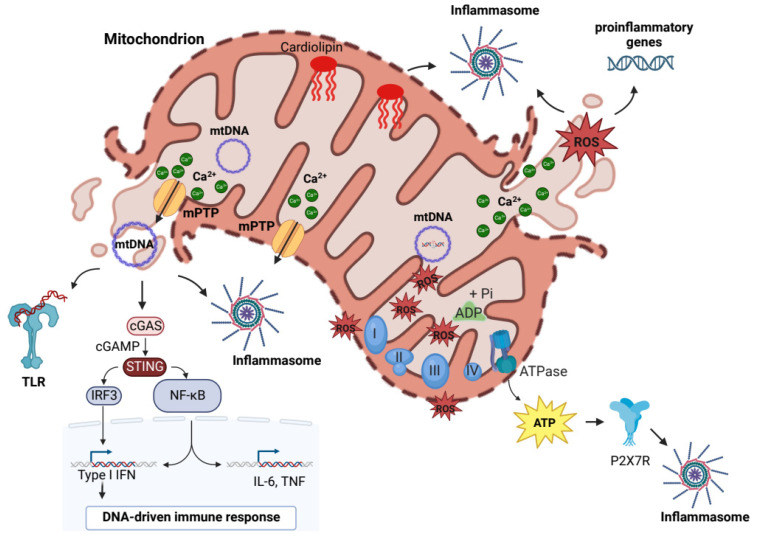
**Several mtDAMPs contribute to mito-inflammation.** The main mtDAMPs involved in mito-inflammation are ATP, cardiolipin, mtROS, Ca^2+^, and mtDNA. ATP-high intracellular levels lead to activation of P2X7 receptor, resulting in activation of NLRP3 inflammasome mediating K^+^-efflux; Cardiolipin modulates NLRP3 inflammasome activation when transferred to OMM or other cellular structures; mtROS at high levels stimulate production of proinflammatory cytokines and promote inflammasome activation; mitochondrial Ca^2+^ perturbation leads to mtROS production and consequent inflammasome activation; mtDNA release from mitochondria, also through mPTP-opening, leads to activation of inflammasomes (NLRP3 and NLRC4), TLR9 signaling, and cGAS-STING pathway. Created with BioRender.com.

**Figure 2 ijms-24-08181-f002:**
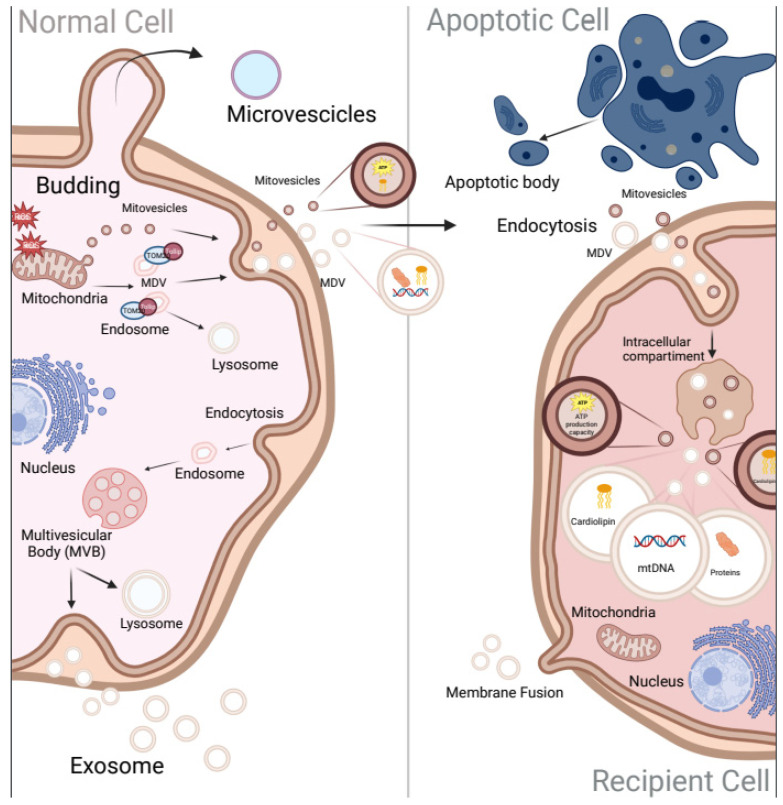
**Extracellular vesicles (EVs) biogenesis.** Extracellular vesicles (EVs) are classified into three types: exosomes, vesicles enclose by the lumen of multivesicular bodies (MVB) that can either ensue the secretory pathway fusing with the PM to discharge its content as exosomes or ensue the lysosome pathway where MVB fuse with the lysosome for degradation; Microvesicles, which generate via budding of the PM; and Apoptotic bodies, PM blebbing by cells undergoing programmed cell death. Under stress conditions, mitochondrial components including mtDNA, ROS, cardiolipin, and proteins have enveloped into TOM20-positive “mitochondrial-derived vesicles” (MDVs). MDVs may be targeted to Tollip-positive endosome facilitating their trafficking to the lysosome or released outside direct to recipient cells. Created with BioRender.com.

**Figure 3 ijms-24-08181-f003:**
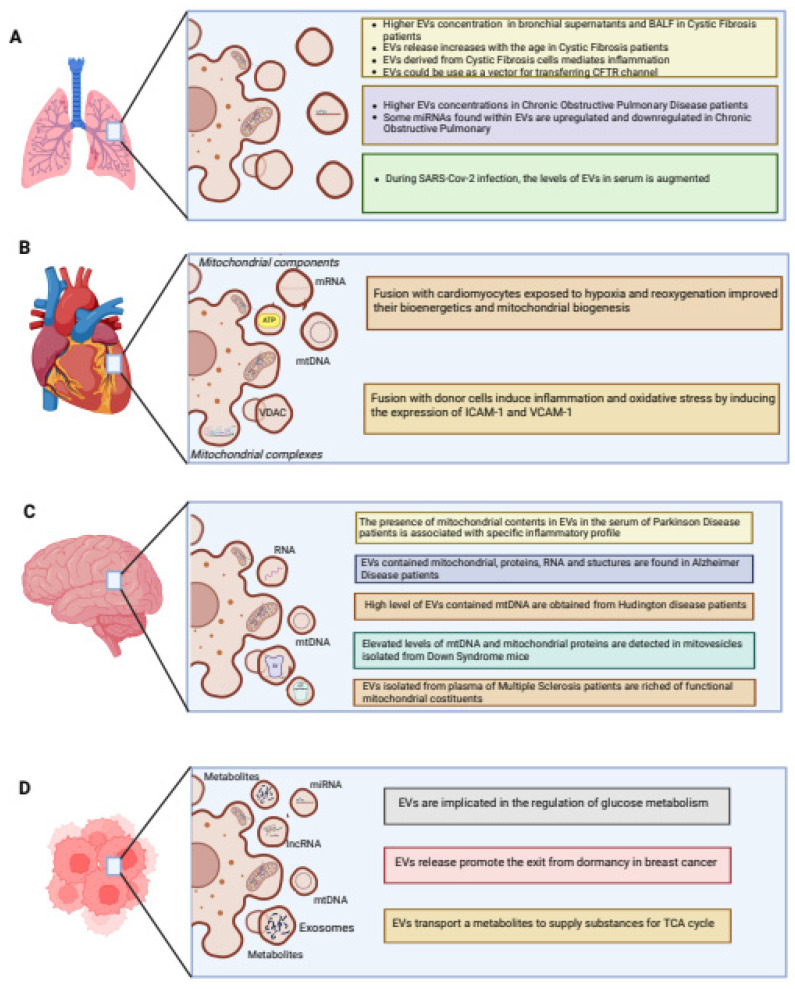
**Extracellular vesicles (EVs) release in inflammatory-related diseases.** (**A**) In the pulmonary system of cystic fibrosis patients, the released EVs are higher respect to control, and this EV release increases with age of cystic fibrosis patients. In Chronic Obstructive Pulmonary Disease (COPD) patients, a higher release of EVs containing miRNAs is observed. In SARS-CoV-2 infection, patients showed an increased EVs release. (**B**) In the cardiovascular system, the EVs are enriched with mRNA, ATP, and mtDNA. The EVs incorporation in cardiomyocytes improved mitochondrial biogenesis during hypoxia and subsequent reoxygenation condition. EVs enriched with VDAC protein and fused with donor cells induced oxidative stress and inflammation by the expression of ICAM-1 and VCAM-1. (**C**) In the nervous system, EVs are involved in different diseases. An increased level of released EVs, containing mtROS, was observed in Parkinson Disease (PD). EVs containing RNA are higher in Alzheimer Disease (AD) patients respect to control. EVs containing mtDNA are increased in Huntington’s disease (HD) and Down Syndrome patients. an increased level of EVs containing a higher mitochondrial complex IV and a lower mitochondrial complex V activity was observed in Multiple Sclerosis (MS). (**D**) In cancers, Evs enriched in miRNAs, metabolites, and lncRNA regulated glucose metabolism. EVs containing mtDNA promoted the exit of breast cancer cells from dormancy. EVs enriched in metabolites supplied substances for TCA cycle. Created with BioRender.com.

## Data Availability

Not applicable.

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
