# Peer review of "The Tricky Connection between Extracellular Vesicles and Mitochondria in Inflammatory-Related Diseases"

_ijms, 2023, doi:10.3390/ijms24098181_

Round 1

Reviewer 1 Report

The review performed by ___ aims to highlight the relationship between EVs containing mitochondrial-derived content and inflammation. This is a very interesting topic and the manuscript is well written and organized. However, some issues should be addressed prior its publication.

1. The text is pretty long, and some sentences can be eliminated as they are not relevant to the main theme of the manuscript. “Similar effects are promoted by the recruitment of Retinoic acid-inducible gene I protein (RIG-I), Melanoma differentiation-associated protein 5 (MDA5) and Mitochondrial Antiviral signaling protein (MAVS), which respond to mtRNA released in the cytosol through mitochondrial permeabilization or via Era like 12S mitochondrial rRNA chaperone 1 (ERAL1).” This sentence can be eliminated. 

2. Some typos must be corrected: “The activation of NLRP3 inflammasome is divide in two phases:” Please correct “divideD”. “In the last few decades, the field of EVs has rapidly expanding revealing the crucial involvement of EVs in a plethora of physiological and pathological processes...” Please correct “expandED”.

3. “CL” please define the first time it is written in the text.

4. TLR9 and TRL9, be careful!

5. Section “Mitochondrial Damage-associated Molecular Patterns” drive the mito-inflammation” should be accompanied by a summary figure to help the reader understand the different mechanisms.

6. Section “The new era of extracellular vesicles communication” can be shortened. Authors should only refer to EVs that contain mitochondrial-derived particles. That means that apoptotic bodies paragraph can be removed. Also, and most importantly, authors should also refer to large EVs that can contain whole and intact mitochondria. 

Author Response

We thank the Editor and the reviewers for the constructive comments, now we have corrected the manuscript following the reviewer's concerns. A new and improved version of manuscript has been submitted; we hope that at this stage the manuscript responds to the standard of IJMS.

REVIEWER 1

The review performed by ___ aims to highlight the relationship between EVs containing mitochondrial-derived content and inflammation. This is a very interesting topic and the manuscript is well written and organized. However, some issues should be addressed prior its publication.

The text is pretty long, and some sentences can be eliminated as they are not relevant to the main theme of the manuscript. “Similar effects are promoted by the recruitment of Retinoic acid-inducible gene I protein (RIG-I), Melanoma differentiation-associated protein 5 (MDA5) and Mitochondrial Antiviral signaling protein (MAVS), which respond to mtRNA released in the cytosol through mitochondrial permeabilization or via Era like 12S mitochondrial rRNA chaperone 1 (ERAL1).” This sentence can be eliminated. 

In accordance with the referee, we have eliminated parts of text that are not relevant to the topics of this manuscript.

Some typos must be corrected: “The activation of NLRP3 inflammasome is divide in two phases:” Please correct “divideD”. “In the last few decades, the field of EVs has rapidly expanding revealing the crucial involvement of EVs in a plethora of physiological and pathological processes...” Please correct “expandED”.

We thank the reviewer; the sentences have been corrected following the referee’s suggestion. However, the manuscript has been revised by a native English-speaking colleague to correct the grammar, spelling, punctuation, and phrasing mistakes.

“CL” please define the first time it is written in the text.

Done.

TLR9 and TRL9, be careful!

We thank the reviewer for the observation, now the mistake has been corrected.

Section “Mitochondrial Damage-associated Molecular Patterns” drive the mito-inflammation” should be accompanied by a summary figure to help the reader understand the different mechanisms.

We agree with the reviewer. In response to this comment, we have added a new figure to review, now figure 1.

Section “The new era of extracellular vesicles communication” can be shortened. Authors should only refer to EVs that contain mitochondrial-derived particles. That means that apoptotic bodies paragraph can be removed. Also, and most importantly, authors should also refer to large EVs that can contain whole and intact mitochondria. 

In agreement with the reviewer, we have revised the section, reducing the introductory part and amplifying the paragraphs referring to EVs containing mitochondria. All modifications reported in the revised text have been marked using the “Track Changes” function.

Reviewer 2 Report

Di Mambro et al. Discuss the role of MDVs and mitovesicles in the inflammatory process under mitochondrial dysfunction and stress conditions that transfer mtDAMPs to recipient cells. This event triggers an anti- or pro-inflammatory response in receiving cells for the regulation of complex biological processes in human diseases. The topic is of great interest to the scientific community and the paper is well organized and the various paragraphs are adequately detailed.

Minor suggestions:

Abstract: add OXPHOS as an important cell process of mitochondria

The acornyms of GAPDH, PKM, ENO1 are missing

“Mitochondrial Damage-associated Molecular Patterns” drive the mito-inflammation paragraph is not exactly correct to say that RET is caused by Ca2+-dependent mitochondrial membrane depolarization. I suggest considering the succinate and NAD concentration, CoQ pool redox state and the membrane potential.

Check “The researchers u sed a high-resolution gradient separation to isolate”

Rephrase the sentence “The new findings raise the intriguing possibility that, in BAK1-competent tumor cells, pharmacological BCL-2- associated X (BAX, apoptosis regulator) inhibitors may accelerate MOMP-driven mtDNA release and potentially offer a window to induce cGAS signaling prior to caspase-dependent cleavage and inactivation of cGAS”

The authors could provide their comments on the role of permeability transition pore in the process of mtDNA release and formation of MDVs or mitovesicles.

Author Response

We thank the Editor and the reviewers for the constructive comments, now we have corrected the manuscript following the reviewer's concerns. A new and improved version of manuscript has been submitted; we hope that at this stage the manuscript responds to the standard of IJMS.  

REVIEWER 2

Di Mambro et al. Discuss the role of MDVs and mitovesicles in the inflammatory process under mitochondrial dysfunction and stress conditions that transfer mtDAMPs to recipient cells. This event triggers an anti- or pro-inflammatory response in receiving cells for the regulation of complex biological processes in human diseases. The topic is of great interest to the scientific community and the paper is well organized and the various paragraphs are adequately detailed.

Minor suggestions:

Abstract: add OXPHOS as an important cell process of mitochondria

Done

The acronyms of GAPDH, PKM, ENO1 are missing

Done.

“Mitochondrial Damage-associated Molecular Patterns” drive the mito-inflammation paragraph is not exactly correct to say that RET is caused by Ca2+-dependent mitochondrial membrane depolarization. I suggest considering the succinate and NAD concentration, CoQ pool redox state and the membrane potential.

The referee is right to point out that this sentence should be clarified. We have extended the sentence as follows: “The levels of mtROS are also linked to the level of mitochondrial Ca2+, which can directly and indirectly stimulate mtROS production: directly, by stimulating mitochondrial resident ROS-generating enzymes, such as ketoglutarate and glycerol 3-phosphate dehydrogenase; indirectly, by the Ca2+-dependent activation of nitric oxide synthase, which mediating nitric oxide blocks the mitochondrial complex IV; and or via reverse electron transport induced by succinate, NAD concentration, CoQ pool redox state and mitochondrial membrane potential”.

Check “The researchers used a high-resolution gradient separation to isolate”

Done.                                           

Rephrase the sentence “The new findings raise the intriguing possibility that, in BAK1-competent tumor cells, pharmacological BCL-2- associated X (BAX, apoptosis regulator) inhibitors may accelerate MOMP-driven mtDNA release and potentially offer a window to induce cGAS signaling prior to caspase-dependent cleavage and inactivation of cGAS”

We have modified the sentences as suggested by the referee as follows: “The new findings raise the possibility that, in BAK1-competent tumor cells, pharmacological BCL-2- associated X (BAX, apoptosis regulator) inhibitors may accelerate MOM permeabilization-driven mtDNA release, inducing cGAS signaling prior to caspase-dependent cleavage and inactivation of cGAS”.

The authors could provide their comments on the role of permeability transition pore in the process of mtDNA release and formation of MDVs or mitovesicles.

In agreement with the reviewer, we have highlighted the role of mPTP in the process of mtDNA release. Regarding the formation of mitovesicles, the role of mPTP is not defined. Evidences show that under stress conditions, and thus when the mitochondria are permeabilized, the level of released mitovesicles are increased. However, no studies have been published until now that investigated the direct role of mtPTP in mitovesicle formation. In the future, it will be interesting to study how, manipulating the expression and functionality of mPTP, changes the mitovesicles formation in different pathological conditions.

Reviewer 3 Report

In the current article by Mambro et al., the authors have tried to address the connection between mitochondrial EVs and different inflammatory diseases. There are numerous issues with the article and hence cannot be accepted for publication.

The Paper is not in format of IJMS.

There are no Line numbers and Page numbers throughout the script so it becomes very difficult for the reviewer to highlight any issue.

The authors need to rectify the English language throughout the script.

The authors have not even numbered the sections and subsections that makes the script very confusing.

The entire article is not focused on the theme as proposed in the title.

The authors start discussing about one point and start getting deeper into the details irrespective of its relevance to the theme of article.

In Abstract:

Paragraph 1: Please check the English in Line 1.

Line 2: The sentence is too lengthy. Please split it into two sentences for a better understanding.

In last line, the authors should keep in mind that it is a review article, not a newspaper article. Please work on the scientific language throughout the script.

In introduction:

The authors have used one paragraph for one line. Was it so special?

The authors have used a term “danger signals”. What are danger signals? Please specify.

In section: “Mitochondrial Damage-associated Molecular Patterns” drive the mito-inflammation

The first line is redundant. Please get rid of that.

Please check the English of second line.

Please check the English of sentence “Consequently, CL is transferred to the OMM or other cellular structures where modulate the activation of inflammatory reactions through activation of the NLRP3 inflammasome”

Please check the typological errors in line “The researchers u sed a high-resolution gradient separation to isolate brain-derived EVs from WT mice to obtain eight fractions containing different-sized EVs carrying various contents.”

The sentence “Specifically, it has been demonstrated that EVs contents mainly consist of DNA, mRNA, miRNA and proteins” is redundant and has been repeated multiple times.

Author Response

We thank the reviewer for the comments, now we have corrected the manuscript following the reviewer's concerns. A new and improved version of manuscript has been submitted; we hope that at this stage the manuscript responds to the standard of IJMS.

REVIEWER 3

In the current article by Mambro et al., the authors have tried to address the connection between mitochondrial EVs and different inflammatory diseases. There are numerous issues with the article and hence cannot be accepted for publication. The Paper is not in format of IJMS.

We are sorry that our manuscript was not appreciated by the referee. Now, the paper has been modified following the IJMS’ standard.

There are no Line numbers and Page numbers throughout the script so it becomes very difficult for the reviewer to highlight any issue.

As requested by the reviewer, we have added line numbers and page numbers.

The authors need to rectify the English language throughout the script.

We thank the reviewer, now, the manuscript has been revised by a native English-speaking colleague to correct the grammar, spelling, punctuation, and phrasing mistakes.

The authors have not even numbered the sections and subsections that makes the script very confusing.

We have numbered all the sections and subsections, as suggested by reviewer.

The entire article is not focused on the theme as proposed in the title. The authors start discussing about one point and start getting deeper into the details irrespective of its relevance to the theme of article.

The review was conceived based on three keywords which have been included in the title: Extracellular vesicles, mitochondria and inflammatory diseases. This manuscript aims to collect all the latest findings linking mitochondria to EVs in the inflammatory process, shedding light on their involvement in several inflammatory-related diseases. For this reason, first, we have introduce the mitochondrion to explain how it is involved in the regulation of the inflammatory processes mediating the release of mtDAMPs. Successively, we have described which subtypes of EVs are known today and we have discussed of their role in several pathological contexts, highlighting how their cargo is mostly made up by mitochondrial portions, mtDNA, mitochondrial proteins and mtDAMPs.

In Abstract:

Paragraph 1: Please check the English in Line 1.

As requested by the reviewer, the sentence has been revised.

Line 2: The sentence is too lengthy. Please split it into two sentences for a better understanding.

Done

In last line, the authors should keep in mind that it is a review article, not a newspaper article. Please work on the scientific language throughout the script.

The abstract has been slightly modified to improve the scientific language. It is not quite clear what the comment on the newspaper article refers to, but we have reflected and tried to edit it with this problem in mind.

In introduction:

The authors have used one paragraph for one line. Was it so special?

We apologize to the referee, but his or her comment is not clear to us. The introduction was written to lay the foundation for the theme of the main body of the review. This is why it was structured with three keywords: mitochondria, mtDAMPs and EVs.

The authors have used a term “danger signals”. What are danger signals? Please specify.

The term 'danger' has been replaced with 'mitochondrial damage-associated signals' and the sentences has been modified to allow an easier understanding of the subject matter.

In section: “Mitochondrial Damage-associated Molecular Patterns” drive the mito-inflammation

The first line is redundant. Please get rid of that.

The first line has been deleted.

Please check the English of second line.

As suggested by the reviewer, The sentence has been corrected.

Please check the English of sentence “Consequently, CL is transferred to the OMM or other cellular structures where modulate the activation of inflammatory reactions through activation of the NLRP3 inflammasome”

As suggested by the reviewer, The sentence has been corrected.

Please check the typological errors in line “The researchers u sed a high-resolution gradient separation to isolate brain-derived EVs from WT mice to obtain eight fractions containing different-sized EVs carrying various contents.”

As suggested by the reviewer, The sentence has been corrected.

The sentence “Specifically, it has been demonstrated that EVs contents mainly consist of DNA, mRNA, miRNA and proteins” is redundant and has been repeated multiple times.

We thank the reviewer for the comment, but checking carefully the text, there are not many repetitions of the sentence. Moreover, the first time that EVs contents is mentioned is in the section 'Mitochondrial Damage-associated Molecular Patterns'. In the different paragraphs in the body of the text, the content of EVs is repeated in relation to the findings made by other researchers on the specific inflammatory-related disease. In order to be less redundant, the sentence “Specifically, it has been demonstrated that EVs contents mainly consist of DNA, mRNA, miRNA and proteins, which provide biological information about the physiological/pathological conditions of the lungs” in the section “Pulmonary system” has been modified.

Round 2

Reviewer 3 Report

The authors have incorporated the suggestions and the script has been improved.